# Molecular Biomarkers for the Early Detection of Ovarian Cancer

**DOI:** 10.3390/ijms231912041

**Published:** 2022-10-10

**Authors:** Ruiqian Zhang, Michelle K. Y. Siu, Hextan Y. S. Ngan, Karen K. L. Chan

**Affiliations:** Departments of Obstetrics and Gynaecology, LKS Faculty of Medicine, The University of Hong Kong, Pok Fu Lam, Hong Kong

**Keywords:** molecular biomarkers, CA125, HE4, RMI, ROMA, ovarian cancer

## Abstract

Ovarian cancer is the deadliest gynecological cancer, leading to over 152,000 deaths each year. A late diagnosis is the primary factor causing a poor prognosis of ovarian cancer and often occurs due to a lack of specific symptoms and effective biomarkers for an early detection. Currently, cancer antigen 125 (CA125) is the most widely used biomarker for ovarian cancer detection, but this approach is limited by a low specificity. In recent years, multimarker panels have been developed by combining molecular biomarkers such as human epididymis secretory protein 4 (HE4), ultrasound results, or menopausal status to improve the diagnostic efficacy. The risk of ovarian malignancy algorithm (ROMA), the risk of malignancy index (RMI), and OVA1 assays have also been clinically used with improved sensitivity and specificity. Ongoing investigations into novel biomarkers such as autoantibodies, ctDNAs, miRNAs, and DNA methylation signatures continue to aim to provide earlier detection methods for ovarian cancer. This paper reviews recent advancements in molecular biomarkers for the early detection of ovarian cancer.

## 1. Introduction

Ovarian cancer is the eighth most common and the fifth deadliest cancer in women worldwide. With an incidence of 3.4% and a mortality of 4.7%, over 300,000 women are afflicted and approximately 152,000 women die of ovarian cancer each year, highlighting the serious threat that this disease poses to the health and survival of women [1]. The prognosis for ovarian cancer patients is poor, with a survival rate of just 30% [2]. The current first-line therapy for ovarian cancer involves a combination of cytoreductive surgery and platinum-based chemotherapy [3]. Targeted therapy, including anti-VEGF antibodies and PARP inhibitors, can be applied to certain patients [4]. However, more than half of the patients will experience a recurrence within two years, resulting in little to no improvement in the survival rate [5,6]. Studies have reported that the 5-year overall survival rate is around 92% for early-stage disease compared with 29% for late-stage disease [7]. Due to the absence of typical signs and symptoms at early disease stages as well as the aggressive tendency of ovarian cancer to progress from an early to advanced stage within 1 year, more than 70% patients are diagnosed at an advanced stage [8]. Therefore, early detection and diagnosis are of great importance to improve the prognosis [9].

Currently, the gold standard for an ovarian cancer diagnosis is based on a histopathology examination [10]. This often requires an operation to remove the tumor and this involves operative risks. In addition, the operations required for the treatment of benign and malignant ovarian lesions differs significantly and, therefore, an accurate preoperative prediction is important. An ovarian tumor is often first detected by transvaginal ultrasound (TVS). A number of ultrasound features have been identified to predict a malignancy, but the diagnostic accuracy still needs to be optimized [11]. A serum biomarker is a convenient, economical, and non-invasive method for predicting the malignancy and efforts have been made to identify biomarkers that are more reliable for an earlier detection of ovarian cancer. 

The search for effective screening methods continues. An acceptable screening method for detecting early-stage ovarian cancer requires a sensitivity greater than 75% and a specificity of at least 99.6% to achieve a positive predictive value (PPV) of at least 10% [12]. The discussion below focuses on recent advances in biomarker developments for the early detection of ovarian cancer, including two current Food and Drug Administration (FDA)-approved markers as well as algorithms or indexes and the ongoing development of potential molecular biomarkers.

## 2. Current Biomarkers for Ovarian Cancer

### 2.1. Cancer Antigen 125 (CA125)

CA125, a glycoprotein encoded by MUC16, is secreted from the coelomic and müllerian epithelia into the bloodstream [13]. CA125 is overexpressed in more than 80% of ovarian cancer patients and can be detected in serum, creating an opportunity to discriminate malignant ovarian tumors from the normal population [14]. In 2011, CA125 was recommended by the National Institute for Health and Care Excellence (NICE) of the UK as a screening test for women with symptoms of possible ovarian cancer [15]. Postmenopausal women with a CA125 level higher than 35 U/mL are considered to have a high risk of a malignancy. As the most studied and most commonly used serum biomarker for an ovarian cancer diagnosis, CA125 is currently the best biomarker. Mukama et al. evaluated the performance of 92 preselected proteins in blood samples collected <18 months prior to an ovarian cancer diagnosis. Other than CA125, the study found no markers that provided diagnostic discriminatory information when ovarian cancer was detected more than 9 months after the blood draw [16].

#### 2.1.1. Sensitivity and Specificity

The accuracy of CA125 for detecting early-stage ovarian cancer is limited; only 50% of early-stage patients have elevated CA125 levels, leading to a low sensitivity (50–62%) for detecting early-stage ovarian cancer. Serum CA125 levels were only able to differentiate advanced-stage patients from healthy controls. Funston et al. retrospectively analyzed CA125 levels before an ovarian cancer diagnosis and reported that patients with normal CA125 levels before the diagnosis were more likely to be diagnosed at an early stage compared with those with elevated CA125 levels. Therefore, screening based only on CA125 may delay the diagnosis and lead to worse outcomes in women [17]. 

Furthermore, the specificity of CA125 is relatively low (generally 73–77%) and more than 60% of patients with increased CA125 levels do not have ovarian cancer [18]. Elevated CA125 levels can be detected due to pregnancy; the menstrual cycle; other malignancies such as breast cancer, uterine cancer, stomach cancer, pancreatic cancer, liver cancer, and colon cancer; and other benign conditions such as acute pelvic inflammation, adenomyosis, uterine myoma, and endometriosis [19]. Our study of 414 Asian women with adnexal masses found that 31.5% of patients with a benign disease had elevated CA125 levels [20]. This leads to a decrease in the PPV; the results from the Prostate, Lung, Colorectal and Ovarian (PLCO) Cancer Screening Trial indicated that the PPV for CA125 alone was only 3.7% [21]. The performance of CA125 varies in different cancer tissue types, with a poor performance in non-epithelial ovarian cancers, clear cell carcinomas, undifferentiated carcinomas, and mucinous carcinomas [22]. Many other factors can influence serum CA125 levels, leading to variations in the baseline CA125 levels in different women. 

#### 2.1.2. Improved Techniques to Detect CA125

To improve the performance of CA125, many strategies have been developed. For example, several studies have developed new techniques to detect serum CA125. Currently, a double determinant immunoassay with an anti-MUC16 antibody (OC125) and an anti-IgM antibody (M11) is used to measure serum CA125 levels. However, one limitation of this method is that these antibodies do not recognize all repeats and may stain other proteins, resulting in a low sensitivity and specificity [23,24]. As such, improved techniques for detecting CA125 have been developed. Wang et al. developed a novel antibody–lectin ELISA assay, which showed an improved specificity for the differential diagnosis of patients with positive CA125 levels, despite having a limited effect on borderline ovarian tumors [25]. The mass spectrometry-based CA125 detection assay developed by Schuster-Little et al. detected the molecular regions that were not recognized by antibodies [26]. Nanoparticles can also be used for CA125 detection by immobilizing CA125 antibodies onto a CuBTC@MoS2-gold nanoparticle (AuNP)-functionalized electrode by electrostatic adsorption to enhance the sensitivity and specificity of CA125 for a diagnosis [27]. Detecting CA125 in exosomes is another method to improve CA125 performance. Detecting exosomal CA125 levels via an immunoassay method provides a better performance compared with traditional serum CA125 detection methods with respect to the area under the curve (AUC) (0.9755 vs. 0.9093), sensitivity (94.55% vs. 87.27%), and specificity (92.73% vs. 90.91%) [28].

#### 2.1.3. Detecting the Glycoforms of CA125

Detecting the glycoforms of CA125 is another strategy to improve the performance of this biomarker. CA125 is a highly glycosylated protein, with more than two-thirds of its molecular weight composed of glycan and an abundance of N- and O-glycans on the extracellular amino terminal domain [29]. The carbohydrate side chains of glycoproteins can be truncated or aberrant in tumor tissue due to different glycosylation processes that occur during the oncogenic transformation [30]. Aberrant N-glycosylation and truncated O-glycans of CA125 have been detected in ovarian cancer, creating opportunities to differentiate ovarian cancer patients from the healthy population [25,31]. Many studies have used CA125 glycoforms for the early detection of ovarian cancer; these improved the specificity and sensitivity compared with the traditional serum protein detection method. 

CA125 has a large amount of Thomsen-nouveau (Tn) antigens (Gal-NAc1-O-Ser/Thr), a type of O-glycan that shows an upregulated expression in ovarian cancer tissue and a low expression in normal cells. Wang et al. detected the combined level of CA125 and Tn (CA125-Tn) using an antibody–lectin (Vicia Villosa Lectin) ELISA assay for an ovarian cancer diagnosis. This combined detection method provided a better performance compared with a traditional CA125 immunoassay, with a significantly increased specificity (75.5% vs. 35.1% specificity at a fixed sensitivity of 90%) among patients older than 45 years. CA125-Tn could also be detected in a low abundance using this method [25]. The sialylation of the Tn structure leads to the formation of the sialyl-Tn antigen (STn), which is found in the mucin-type glycoproteins of most types of human adenocarcinomas and is limited in normal cells. The total serum concentrations of the STn antigen have been found by a radioimmunoassay to increase by 50%, 9.6%, and 3.8% in ovarian cancer patients, endometriosis patients, and healthy controls, respectively. Different methods have been reported to detect CA125-STn, including a time-resolved fluorometry immunoassay [32], a glycovariant-based lateral flow immunoassay (LFIA) [33], and fluorescent europium nanoparticles coated with anti-STn-mAbs [34,35], all of which performed better than the conventional CA125 immunoassays by increasing the sensitivity and reducing the false-positive rates. The most significant performance improvements have been seen in postmenopausal cases and in patients with marginally elevated serum CA125 levels, providing an opportunity for the improvement of early ovarian cancer detection at very low marker concentrations. However, the CA125-STn approach remains limited for the detection of several ovarian cancer histologies such as clear cell and mucinous cancers.

#### 2.1.4. Risk of Ovarian Cancer Algorithm (ROCA)

CA125 has a long biological half-life, indicating that the continuous measurement of serum CA125 levels may be advantageous [36]. The ROCA is an assay used to calculate the risk of ovarian cancer based on serial CA125 serum measurements [37]. This assay aims to monitor significant increases in CA125 levels, such that people with strongly elevated CA125 levels can be further assessed by TVS [38]. The combined application of the ROCA and TVS increases the limited specificity of the single-threshold CA125 test, improving the sensitivity to 85% for an earlier detection [39]. With a longitudinal CA125 method of mean trends (MMT) assay in 360 postmenopausal women, the AUC was 0.911 and the sensitivity was 90.5% [37]. The UK Collaborative Trial of Ovarian Cancer Screening (UKCTOCS) conducted studies of annual multimodal screening (longitudinal CA125 and second-line TVS) as well as annual TVS screenings (TVS first- and second-line tests) and found that the CA125 test screen allowed for a subtle stage shift, with stage I and stage IV cancer incidence 47.2% higher and 24.5% lower, respectively, compared with an unscreened group. However, this stage shift was too small to yield a statistically significant decrease in mortality after a long-term follow-up (median follow-up > 16 years after recruitment) [40].

### 2.2. Human Epididymis Secretory Protein 4 (HE4)

HE4 is a member of the whey acidic four-disulfide core (WFDC) protein family that was originally identified in the epithelium of the distal epididymis [41]. It is a peptide protease inhibitor involved in the innate immune response of epithelial tissues [42,43]. HE4 is not found in the ovarian surface epithelium; however, it is overexpressed in ovarian cancer tissue, where it is secreted into the extracellular environment and can be detected in the blood stream [44]. Therefore, the detection of serum HE4 is another potential biomarker for the diagnosis and monitoring of ovarian cancer. 

#### 2.2.1. Sensitivity and Specificity

Studies have reported that measuring the HE4 levels provides an ability to detect ovarian cancer with a specificity of 96% and a sensitivity of 67% [45]. Compared with CA125, HE4 is less frequently affected by benign gynecological conditions; it is not elevated in endometriosis and it has only been found to increase in adenomyosis patients [46]. Furthermore, Chan et al. found that HE4 showed a better sensitivity in mucinous tumors but was not strongly expressed in clear cell carcinomas [20]. HE4 was found to be elevated in more than half of ovarian tumors that did not express CA125. HE4 is not specific to ovarian cancer; it is also highly expressed in endometrial cancer, lung adenocarcinomas, squamous cell carcinomas, breast adenocarcinomas, and mesotheliomas. In early ovarian cancer, the pooled sensitivity for HE4 detection was 0.64 and the specificity was 0.87 [47]. In late ovarian cancer, the pooled sensitivity was 0.89 and the specificity was 0.86 [48]. CA125 has been reported to give a better sensitivity than HE4 in a late-stage disease (90.8% and 56.9%, respectively), but HE4 performed better than CA125 with respect to the specificity (96.9% vs. 67.1%) and PPV (78.7% vs. 35.8%) [49]. In a systematic review of 49 previous articles that studied the diagnostic role of HE4, including 12,631 women and 4549 ovarian cancer patients, HE4 had a pooled sensitivity of 0.78 and a specificity of 0.86 for the detection of borderline or malignant ovarian tumors [50].

#### 2.2.2. Factors Affecting HE4 Levels

HE4 levels are influenced by the menopausal status and age. HE4 detection performs better in postmenopausal women than in premenopausal women, with a sensitivity of 77% and 71%, respectively, and a specificity of 91% and 88%, respectively [48]. HE4 levels increase with age, which leads to a decreased specificity and sensitivity in the older population [51]. Previous studies have compared the performance of HE4 in women under 50 years of age with that of women over 50 years of age and found that the sensitivity decreased from 100% to 87.5% and the specificity decreased from 88.4% to 60.4% in the older group [50]. 

## 3. Current Multivariate Index Assays for Ovarian Cancer

### 3.1. Risk of Malignancy Index (RMI) Assay

Due to the limited efficacy of single serum biomarkers, many researchers have tried to combine several indexes to improve performance biomarker applications. In 1990, Jacobs et al. established the RMI by multiplying ultrasound results (U), CA125 levels, and menopausal status (M) to predict the risk of an ovarian malignancy (RMI = U × M × CA125) [52]. Using a cut-off value of 200, the RMI demonstrated an increased sensitivity (71–88%) and specificity (74–92%) compared with an assessment of the CA125 levels alone [53]. 

Later, Tingulstad et al. developed RMI 2 (sensitivity of 71% and specificity of 96%) and RMI 3 (sensitivity of 71% and specificity of 92%) and Yamamoto et al. developed RMI 4 (sensitivity of 86.8% and specificity of 91%). These three new versions employ a modified scoring of the U and M parameters. In addition, RMI 4 also takes the tumor size into consideration [54,55,56]. The cut-off values with the best differentiation points were 200 for RMI 1–3 and 450 for RMI 4. 

Many studies have been conducted to compare the efficacy of these four RMI versions. A systematic review has reported that RMI 1 had the highest accuracy among RMI 1–3 (*53*). The NICE guidelines based on this study recommended the use of the RMI 1 score to manage suspected ovarian malignancies [57]. However, a recent study has shown that, despite a higher sensitivity in RMI 2 and 4 as well as a higher specificity in RMI 1, no significant differences in the AUC were found among them [58]. 

### 3.2. OVA1 Assay

OVA1 is an FDA-approved multivariate index assay that incorporates the serum biomarkers CA125, transthyretin, transferrin, beta-2 microglobulin, and apolipoprotein A-1, aiming to calculate the risk index score of an ovarian malignancy. The performance of OVA1 is better than that of the CA125 detection alone with respect to the sensitivity (92% vs. 79%, respectively) and the negative predictive value (NPV) (97% vs. 93%, respectively) [59]. Furthermore, in 63–78% of early-stage ovarian cancer patients and 50–58% of patients with less frequently diagnosed histology subtypes (including clear cell carcinomas, mucinous carcinomas, and sex-cord stromal tumors), those with a low CA125 level could be identified by OVCA1 [60,61,62]. Therefore, OVA1 is able to identify ovarian cancer patients that would otherwise be missed by CA125 screening, thereby leading to an early detection and a better prognosis.

### 3.3. Risk of Ovarian Malignancy Algorithm (ROMA) Assay

Moore et al. established the multivariate index ROMA by incorporating the CA125 serum level, HE4, and menopausal status using a logistic regression model. The ROMA index was approved by the FDA for ovarian cancer diagnoses in 2010 [46,63] and it provides a better predictive value than CA125 or HE4 detection alone [64]. A meta-analysis retrospectively evaluated the ROMA index for 5954 cases. The pooled estimates for the ROMA index showed a sensitivity of 90%, a specificity of 91%, an AUC of 0.96, a PPV of 90%, and an NPV of 93%, indicating that the ROMA index provides a reliable basis for the clinical diagnosis of ovarian cancer [65].

Chan et al. found that the ROMA had an 83% accuracy in diagnosing an early-stage disease [20] and its sensitivity was higher in postmenopausal women than in premenopausal women (82.5–90.8% vs. 53.3–72.7%, respectively). However, the ROMA index showed a lower specificity in postmenopausal women than in premenopausal women (66.3–84.6% vs. 74.2–87.9%, respectively). A meta-analysis (32 studies) conducted by Suri et al. indicated that the ROMA index performed better with respect to the diagnostic accuracy (highest DOR, sensitivity, and AUC) compared with the detection of CA125 or HE4 alone in postmenopausal women; however, in premenopausal women, HE4 showed the highest specificity, AUC, and DOR [66]. The ROMA showed a similar sensitivity but an improved specificity when compared with CA125, especially in premenopausal women. Chan et al. further found that the ROMA showed an improved specificity and PPV (34.69% vs. 16.8%, respectively) but a similar specificity and PPV when compared with CA125 for the prediction of ovarian cancer in Asian women with a pelvic mass. The ROMA also showed a significantly higher sensitivity (89.2%) compared with the RMI [20,67]; however, another study reported that the ROMA did not offer any added clinical benefit over CA125 or HE4 detection [68].

Using the OVA1 and ROMA tests sequentially may enhance the PPV by exploiting the high sensitivity of OVA1 in the first-line test, followed by the high specificity of the ROMA for all women with high OVA1 scores in the second-line test [69]. 

### 3.4. IOTA Simple Rules and the ADNEX Model

The International Ovarian Tumor Analysis (IOTA) simple rules were developed based on an ultrasound examination with a 92% sensitivity and a 96% specificity. Chan et al. evaluated the IOTA with a subjective assessment using expert ultrasound, the RMI, and the ROMA for assessing the nature of a pelvic mass [70]. Chan et al. investigated whether the ROMA could replace expert ultrasound when the IOTA results were inconclusive and showed that expert ultrasound was more sensitive than the ROMA for diagnosing an ovarian malignancy in such cases without significant differences in the specificity or accuracy [70]. Furthermore, similar accuracies were found in all of the assessment methods involving the IOTA, which were more accurate than the RMI or ROMA alone. Therefore, the IOTA should be the first method used to assess a pelvic mass. If inconclusive, an assessment by expert ultrasound is preferable [70].

The IOTA group also developed the Assessment of Different Neoplasias in the Iexa (ADNEX) model by combining six ultrasound features and three clinical features (age, serum CA125 level, and type of center). This model could be used to indicate a specific malignancy subtype [71]. A 17-center cohort study indicated that the IOTA ADNEX model outperformed the first two of the six models (RMI, logistic regression model 2, Simple Rules, Simple Rules risk model, and ADNEX model with or without CA125), with a sensitivity of 86.5% at a 90% specificity and a specificity of 86.6% at a 90% sensitivity for assessing pelvic masses based on the results for 4905 patients [72]. 

## 4. Potential Biomarkers for Ovarian Cancer Detection

### 4.1. Potential Protein Biomarkers for Ovarian Cancer Detection

Protein biomarkers have been widely studied during the past 3 decades and more than 100 potential biomarkers have been evaluated. Folate receptor alpha (FOLR1) is a membrane protein regulating the binding and cellular uptake of folic acid into cells [73]. The FOLR1 expression is restricted to the luminal surfaces of the epithelial cells in healthy populations, but it is highly expressed in many epithelial cancers, including breast cancer, ovarian cancer, clear cell renal carcinomas, endometrial carcinomas, and lung cancer [74]. Around 76% of high-grade ovarian cancer patients show a FOLR1 overexpression [75]. In addition, FOLR1 can be secreted into the serum in a soluble form via GPI-specific serum phospholipase or membrane-associated protease. The serum FOLR1 level was significantly elevated in ovarian cancer patients compared with healthy and benign tumor populations [76]. Serum FOLR1 has also shown an increased specificity compared with CA125, which has demonstrated a better diagnostic performance [77]. Another limitation is that the level of FOLR1 is affected by the tumor histotype, clinical grade, stage, and tumor size. Most patients with an elevated FOLR1 level have tumors of a serous subtype and are at a late disease stage. The FOLR1 level is much lower in mucinous tumors and in early tumor stages [77].

CA72-4 is a tumor-associated glycoprotein. It is a distinct epitope on the MUC1 mucin and its abnormal elevation has been detected in ovarian cancer [78]. Its level is not influenced by pregnancy, the menstrual cycle, or endometriosis [79,80] and is only slightly affected by inflammatory conditions [81]. Therefore, the addition of CA72-4 to CA125 could increase the diagnostic specificity, but at the cost of the sensitivity [57]. Furthermore, its overexpression has been detected in many ovarian clear cell carcinomas and mucinous tumor cases whereas CA125 and HE4 levels are generally not elevated in these two histotypes, which means that CA72-4 may have the potential to detect cases missed by CA125 and HE4 [82,83]. However, the sensitivity of CA72-4 as a single marker is limited [84]. 

Transthyretin (TTR) is another potential biomarker that is downregulated in ovarian cancer patients. Zheng et al. reported that TTR performed better than CA125 and HE4 in the detection of early-stage (stages I and II) ovarian cancer [85]. However, due to the low sensitivity, current studies have mainly focused on combining TTR with other biomarkers. Kozak et al. combined TTR with CA125, hemoglobin, apolipoprotein AI, and transferrin. This combination might be of benefit in the detection of early ovarian cancer [86].

Other molecular biomarkers for the detection of early ovarian cancer are under investigation, including CA15-3 [78], glycodelin [87], and kallikrein 11 [88]. Despite the identification of a variety of new potential biomarkers, none of them outperformed CA125. However, the new biomarkers can be used in combination with CA125 to achieve a better diagnostic performance.

### 4.2. Potential Multivariate Index Assays for Ovarian Cancer Detection

Karlsen et al. established the Copenhagen Index (CPH-I) in 2015 based on age along with the serum levels of HE4 and CA125 [89]. This model has been tested to have a sensitivity of 69% and a specificity of 85% for the differentiation of malignant and borderline tumors [89,90]. Previous studies have indicated that CPH-I has a higher or similar sensitivity but a lower specificity compared with the ROMA [91,92]. The Risk of Ovarian Malignancy Index (ROMI) is a multivariate index assay that is based on the serum levels of CA125, HE4, and TK1. A higher accuracy, sensitivity, specificity, AUC, NPV, and PPV have been reported with the ROMI compared with the ROMA in both pre- and postmenopausal patients [93]. A multivariable model based on five proteins (CA125, OPN, HE4, leptin, and prolactin) performed better than CA125 alone with respect to the AUC (0.996 vs. 0.929) [94]. A combined panel of annexin A2 (ANXA2) and CA125 showed a sensitivity of 100%, a specificity of 63.6%, and an accuracy of 71.4% for distinguishing stage IA ovarian cancer from benign ovarian lesions, which was more accurate than the CA125 detection alone [95]. A combined model of insulin-like growth factor-binding protein-2 (IGFBP-2), lecithin cholesterol acyltransferase (LCAT), and CA125 outperformed CA125 detection alone for the earlier detection of ovarian cancer in terms of the sensitivity. This combined model also increased the lead time by 5–6 months [96]. A longitudinal multimarker model of CA125, arginase 2 (AGR2), and chitinase-3-like protein 1 (CHI3L1) showed a significant improvement in the sensitivity and a lead time of 1–2 years for the diagnosis compared with the use of CA125 alone [97].

### 4.3. Potential Role of Autoantibodies (AABs) in the Early Detection of Ovarian Cancer

The genetic alteration of cancer cells leads to an aberrant expression of tumor-associated antigens (TAA) that can be recognized by the immune system, resulting in the generation of corresponding AABs [98]. AABs are stable proteins that can be detected in the circulation for long periods of time and typically have higher concentrations due to an immune system-induced amplification, such that they can detect aberrant antigens at low concentrations [99]. AABs provide a new insight into cancer detection. 

The anti-TP53 autoantibody is the best-studied for ovarian cancer detection. The tumor suppressor gene TP53 is mutated in more than 95% of high-grade serous ovarian cancer patients [100]. Due to the fact that TP53 mutations occur in the early stage during carcinogenesis, the lead time of TP53 AABs to diagnosis ovarian cancer is longer than with traditional detection methods, with an average elevation of 8.1 months and 9.2 months compared with a detection by CA125 alone or the ROCA, respectively [101]. Of these patients, three out of four had false-negative CA125 levels. However, the sensitivity of anti-TP53 autoantibodies is limited; the presence of anti-TP53 AABs could be detected in only ~20% of patients when diagnosed, with this percentage slightly rising (40%) in high-grade serous ovarian cancer (HGSOC) subtype patients. The sensitivity of AABs to TP53 at a specificity of 98% was close to zero. Therefore, because using the anti-TP53 AAB alone showed a limited effect for the early diagnosis of ovarian cancer, many studies have tried to use an optimized panel combining several AABs. Ma et al. screened 154 AABs and reported that an optimized panel of anti-TP53, anti-guanine nucleotide-binding protein (GNAS), and anti-nucleophosmin 1 (NPM1) AABs could discriminate ovarian cancer patients from healthy patients with a sensitivity of 51.2%, a specificity of 86.0%, and an accuracy of 68.6% [102]. The combination of nine AABs (against p53, c-MYC, p90, p62, alpha 2-HS glycoprotein (AHSG), 14-3-3 zeta, RAS-like proto-oncogene A (RalA), KH domain-containing protein overexpressed in cancer (Koc), and P16) showed a sensitivity of 61.4% at a specificity of 85.0% in detecting ovarian cancer [103]. Another panel (including AABs against survivin, p53, p16, cyclin B1, cyclin D1, cyclin A, cyclin E, Koc, IGF2 mRNA-binding protein 1 (IMP1), P62, cyclin-dependent kinase 2 (CDK2), P90, and c-MYC) achieved a sensitivity of 62.5% and a specificity of 85% in the detection of ovarian cancer [104]. Additionally, combining anti-TP53 AABs and CA125 levels was shown to increase the AUC from 0.751 to 0.861 compared with CA125 detection alone [101].

Studies on other AABs have been conducted and several have been validated to achieve good effects for ovarian cancer detection. For example, combining a panel of two AABs (anti-leucine repeat death domain-containing protein (LRDD) and anti-forkhead box A1 (FOXA1) autoantibodies) with serum CA125 levels increased the diagnostic performance by raising the positive rate from 62.7% to 87.1% in ovarian cancer patients [105]. Combining anti-BRCA1-associated RING domain 1 (BARD1) AABs and serum CA125 levels showed a sensitivity of 91% and a specificity of 95% among 741 samples [106]. Lastly, combining anti-PDZ and LIM domain 1 (PDLIM1) AABs and CA125 levels increased the AUC to 0.846, such that around 80% of patients could be detected [107]. 

### 4.4. Potential Role of Circulating Tumor DNA (ctDNA) in the Early Detection of Ovarian Cancer

ctDNAs are DNA fragments that are released from cancer tissues into circulating bodily fluids such as blood, urine, and ascites through apoptosis, necrosis, lysis, and active secretion [108]. ctDNAs can be detected and quantified using PCR, BEAMing technology, and sequencing. Cancer tissues are characterized by specific genetic alterations such as point mutations, copy number alterations, deletions, and epigenetic alterations. Studies have identified that these tumor-related genetic changes are also present in ctDNAs, even in patients at early stages of the disease. Furthermore, the mutation profiles are concordant between solid tumors and matched ctDNAs, which indicates the potential application of ctDNA to detect ovarian cancer in a non-invasive way [109]. Moreover, the half-life of ctDNA is short at around 1 h, which confers the ability to monitor the real-time tumor progression.

In a meta-analysis of 22 studies, the DNA content greatly differed between the serum and plasma. The ctDNA in plasma was less diagnosable than that in serum, with AUCs of 0.88 and 0.92, respectively [110,111]. In contrast, Morgan et al. reported that ctDNA was more diagnosable in plasma, possibly due to the fact that ctDNA is released through leukocyte lysis in the clotting process and is exogenous genomic DNA, which would decrease the proportion of ctDNA [112]. However, the ctDNA level is associated with the tumor burden; therefore, using ctDNA to diagnosis patients at early stages might be challenging. This is especially true in tumors with a diameter of less than 10 mm, in which the ctDNA levels may be too low to be detected. This characterization of ctDNA could result in a low sensitivity for the early detection of ovarian cancer. Additionally, the ctDNA levels may be increased in several physiological conditions such as inflammation, exercise, injury, and surgery, which would lower the specificity for the cancer diagnosis. The heterogeneity of ovarian cancer may pose another challenge for using ctDNA as a biomarker for an ovarian cancer diagnosis. Li et al. conducted a meta-analysis based on 22 previous studies, which included a total of 1125 patients and 1244 healthy controls, and found that the circulating ctDNA showed a comparable performance with the CA125 and HE4 biomarkers, with AUCs of 0.8958, 0.883, and 0.899, respectively [110].

### 4.5. Potential Role of Methylation in the Early Detection of Ovarian Cancer

During cancer development and progression, the hypermethylation of CpG islands in the gene promoter is a frequent event that leads to the repression of transcription, the silencing of tumor suppressor genes, and the activation of oncogenes, ultimately promoting the cancer transformation [113]. Aberrant methylation events occur early in cancer, and are among the earliest molecular changes during carcinogenesis. Methylated DNA is stable both chemically and biologically. Aberrant DNA methylation can be analyzed by methylation-specific PCR (MS-PCR), MethDet assays, MethyLight assays, clonal bisulfite sequencing, or microarray-mediated methylation assays (M^3^-assays) [114,115]. Singh et al. used a quantitative TaqMan-based qPCR assay (MethyLight) and a clonal bisulfite sequencing method to analyze the DNA methylation status of ovarian cancer in the frequently methylated tumor suppressor genes HOXA9 and HIC1. The combination panel showed a sensitivity of 88.9%, a specificity of 100%, and an AUC of 0.95 to discriminate ovarian cancer from healthy patients [114,116]. The combined use of this panel with CA125 further increased the diagnostic accuracy [116]. Li et al. analyzed 22 previous studies and reported that the methylation test showed a better performance than the ctDNA concentration test with respect to the AUC (0.93 vs. 0.9) for the prediction of ovarian cancer [110].

### 4.6. Potential Role of microRNA (miRNA) in the Early Detection of Ovarian Cancer

MiRNAs are single-stranded short non-coding RNAs between 19 and 25 nucleotides in length that bind to target mRNAs to regulate their expression [117]. miRNAs have been shown to regulate the expression of many genes, including those that are aberrantly expressed in cancer cells as well as those that are known to promote carcinogenic processes such as cell proliferation, differentiation, and apoptosis [118]. Given the differential miRNA expression patterns that have been identified between cancer and healthy patients, miRNAs have the potential to be used as biomarkers for cancer detection. 

Teng et al. completed a meta-analysis of five independent clinical studies to determine the diagnostic value of measuring miR-200a-3p and miR-200c-3p levels. Their study found that miR-200a-3p had a relatively high sensitivity of 84%, a specificity of 83%, and a summary AUC of 0.89; miR-200c-3p had a similar sensitivity of 75%, a specificity of 66%, and a summary AUC of 0.77. Both miRNAs showed a relatively high diagnostic efficacy for ovarian cancer [119]. The level of miR-205 has been reported to be elevated in cancer patients and has been shown to have the potential for distinguishing cancer patients from healthy people; miR-205 was shown to have an AUC of 0.715, a sensitivity of 66.7%, and a specificity of 78.1%. A combination detection panel using miR-205, CA125, and HE4 showed an increased AUC of 0.951 and a sensitivity and specificity of 100% and 86.1%, respectively, and performed best in early detection [120]. 

## 5. Conclusions

Identifying more accurate molecular biomarkers for the early detection of ovarian cancer remains an important unmet medical need. CA125 is still the best and most widely used biomarker for the early detection of ovarian cancer in clinics, but it is limited by a low specificity (Table 1). Multivariate panels using other biomarkers to complement CA125 have been shown to have an improved diagnostic performance compared with CA125 alone and a multivariate index panel with the RMI, OVA1, and ROMA has been approved by the FDA for use in clinics (Table 1). Additionally, many studies have identified the potential of other molecular biomarkers for detecting ovarian cancer at early stages, including AABs, ctDNA, methylation, and miRNAs. Further studies are needed to identify the biomarkers with the highest detection and lead times to maximize the survival rate and prognosis for ovarian cancer patients. 

## Figures and Tables

**Table 1 ijms-23-12041-t001:** Diagnostic performance of clinically used molecular biomarkers in the subset of studies cited in this article.

Ref.	Systematic Review or Meta-Analysis	Sample Size	CA125	HE4	ROMA	RMI
Patients	Control	Se (%) (95% CI)	Sp (%) (95% CI)	PPV (%)	NPV (%)	AUC (95% CI)	Se (%) (95% CI)	Sp (%) (95% CI)	PPV (%)	NPV (%)	AUC (95% CI)	Se (%) (95% CI)	Sp (%) (95% CI)	PPV (%)	NPV (%)	AUC (95% CI)	Se (%) (95% CI)	Sp (%) (95% CI)	PPV (%)	NPV (%)	AUC (95% CI)
[16]	No	91	182					0.77															
[20]	No	57	271	90.8	67.1	35.8	97.3		56.9	96.9	78.7	91.8		89.2	87.3	58.6	97.6						
[65]	No	50	100	52	54	54			70	86	74			84	72	84							
[46]	No	70	762	71.4	74.8	20.7	96.6	0.811					0.896										
[48]	Yes	4549	8082						78	86													
[50]	No	82	1147	80.5	92.2			0.927	90.2	75.6			0.927	87.8	80.8			0.959					
[65]	Yes	2117	3837											90	91	90	93	0.96					
[66]	Yes			84	73			0.86	73	90			0.91	86	79			0.91					
[67]	No	56	225	87.9	46.2	29.8	93.6	0.81	53.4	97.8	86.1	89	0.77	79.3	79.8	50.5	93.7	0.89	64	89.8	61	91	
[69]	No	31	115											83.9	83.5	57.8	95.1						
[70]	No	179	511											74.3	84.4				66.5	91.1			
[90]	No	157	110											71	88	90	68	0.88

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
