# Peer review of "Molecular Biomarkers for the Early Detection of Ovarian Cancer"

_ijms, 2022, doi:10.3390/ijms231912041_

Round 1

Reviewer 1 Report

The search for suitable biomarkers for the early detection of ovarian cancer remains a research area of immense importance as ovarian cancer is a lethal cancer with very poor survival rates. This review is timely as previous reviews on this subject were ten years ago; so the need to provide updated information on biomarkers for ovarian cancer is very pressing. Currently, only two markers [CA125 and Human Epididymis protein 4 (HE4)] have been approved by the FDA for monitoring treatment and detecting disease recurrence. Previous reviews have identified a number of potential biomarkers, some of which are not discussed in this current review. Interestingly.  folate receptor-alpha (FOLR1);  which has been extensively portrayed as a promising biomarker was not mentioned in this review. This candidate was subjected to clinical trials previously. I would recommend that this review be updated with more potential markers /what may have led to those biomarkers being considered less suitable. This is necessary to provide an up -to-date information for researchers in this field of study. 

Author Response

Reviewer #1:

Comment 1:

The search for suitable biomarkers for the early detection of ovarian cancer remains a research area of immense importance as ovarian cancer is a lethal cancer with very poor survival rates. This review is timely as previous reviews on this subject were ten years ago; so the need to provide updated information on biomarkers for ovarian cancer is very pressing. Currently, only two markers [CA125 and Human Epididymis protein 4 (HE4)] have been approved by the FDA for monitoring treatment and detecting disease recurrence. Previous reviews have identified a number of potential biomarkers, some of which are not discussed in this current review. Interestingly.  folate receptor-alpha (FOLR1);  which has been extensively portrayed as a promising biomarker was not mentioned in this review. This candidate was subjected to clinical trials previously.

Response to comment 1:

We appreciate the thoughtful comments of the reviewer. We have added information on folate receptor-alpha (FOLR1) as a potential biomarker for ovarian cancer in Section 4.1. Potential protein biomarkers for ovarian cancer detection on page 6 as follows.

“Protein biomarkers have been widely studied during the past three decades and more than 100 potential biomarkers have been evaluated. Folate receptor alpha (FOLR1) is a membrane protein regulating binding and cellular uptake of folic acid into cells (74). FOLR1 expression is restricted to luminal surfaces of epithelial cells in healthy populations, but is highly expressed in many epithelial cancers, including breast cancer, ovarian cancer, clear cell renal carcinoma, endometrial carcinoma, and lung cancer (75). Around 76% of high grade ovarian cancer patients show FOLR1 overexpression (76). In addition, FOLR1 can be secreted into the serum in a soluble form via GPI-specific serum phospholipase or membrane-associated protease. Serum FOLR1 level is significantly elevated in ovarian cancer patients compared to healthy and benign tumor populations (77). Serum FOLR1 has also shown an increased specificity compared to CA125, which has demonstrated a better diagnostic performance (78). Another limitation is that the level of FOLR1 is affected by tumor histotype, clinical grade, stage, and tumor size. Most patients with elevated FOLR1 level have tumors of a serous subtype and are at a late disease stage. The FOLR1 level is much lower in mucinous tumors and in early tumor stages (78).”

Comment 2:

I would recommend that this review be updated with more potential markers /what may have led to those biomarkers being considered less suitable. This is necessary to provide an up -to-date information for researchers in this field of study. 

Response to comment 2:

We have added more information on potential markers in Section 4.1. Potential protein biomarkers for ovarian cancer detection on pages 6-7 as follows.

“CA72-4 is a tumor-associated glycoprotein. It is a distinct epitope on the MUC1 mucin, and its abnormal elevation have been detected in ovarian cancer (79). Its level is not influenced by pregnancy, menstrual cycle, and endometriosis (80, 81) and is only slightly affected by inflammatory conditions (82). Therefore, addition of CA72-4 to CA125 could increase diagnostic specificity, but at the cost of sensitivity (58). Furthermore, its overexpression has been detected in many ovarian clear cell carcinoma and mucinous tumor cases, whereas CA125 and HE4 levels are generally not elevated in these two histotypes, which means that CA72-4 may have the potential to detect cases missed by CA125 and HE4 (83, 84). However, the sensitivity of CA72-4 as a single marker is limited (85).

Transthyretin (TTR) is another potential biomarker that is downregulated in ovarian cancer patients. Zheng et al. have reported that TTR performed better than CA125 and HE4 in the detection of early-stage (stages I and II) ovarian cancer (86). However, due to low sensitivity, current studies have mainly focused on combining TTR with other biomarkers. Kozak et al. have combined TTR with CA125, hemoglobin, apolipoprotein AI, and transferrin. This combination might be of benefit in early ovarian cancer detection (87).

Other molecular biomarkers for early ovarian cancer detection are under investigation, including CA15-3 (79), glycodelin (88), and kallikrein 11 (89). Despite the identification of a variety of new potential biomarkers, none of them outperformed CA125. However, the new biomarkers can be used in combination with CA125 to achieve a better diagnostic performance.”

Comment 3:

Moderate English changes required.

Response to comment 3:

The English editing is completed and the certificate is attached.

Reviewer 2 Report

The work provided a comprehensive summarizes of recent progresses in biomarker development for ovarian cancer early detection, including two common clinic molecular markers, current algorithms or indexes, and ongoing developments for novel biomarkers. The references are adequate and they are well summarized in this work.

Several suggestions listed below may help improve this review.

1)    Line 48-50, The statements of screening method for ovarian cancer, the authors didn’t summarize the cited work correctly. In cited work (12), the sensitivity and specificity were designed to achieve a PPV of 10% which is based on the prevalence of ovarian cancer (1 in 2500 postmenopausal women).

2)    In the section 1 introduction, a brief summary at the end about the motivation of this work can improve the work.

3)    Line 65. ‘Useful’ can be replaced by ‘diagnostic’.

4)    Line 68. ‘usefulness’ can be replaced by ‘sensitivity or accuracy’.

5)    In section 2.2, line 158, the statement of HE4 it not complete or even not accurate. Please check works such as https://www.mdpi.com/1422-0067/14/11/22655/htm and https://www.ncbi.nlm.nih.gov/pmc/articles/PMC5928211/. In a brief, Human epididymis protein 4 (HE4) was first identified in the epithelium of the distal epididymis, belonged to the family of whey acidic four-disulfide core (WFDC) proteins. HE4 is a endogenous peptide protease inhibitors. HE4 is also found to be involved in the innate immune response of epithelial tissues, including ovarian epithelial cells.

6)    This work is a review article, however, authors incorrectly used the phrase ‘we found’ as line 170, 221, or ‘our study’ as line 230. Authors need to rephrase them appropriately.

7)    In section 3.1, authors only mentioned RMI briefly. However, several improved versions of RMI such as RMI2, 3 and 4 have been reported and used in the clinic https://ovarianresearch.biomedcentral.com/articles/10.1186/s13048-020-00643-6. It would be more instructive if authors can discuss them more deeply and with more details.

8)    At the conclusion section, line 391, identifying biomarkers is not a problem. Authors need to rephrase it more correctly, such as ‘unmet medical need or goal’.

Author Response

Reviewer2#:

Comment 1:

Line 48-50, The statements of screening method for ovarian cancer, the authors didn’t summarize the cited work correctly. In cited work (12), the sensitivity and specificity were designed to achieve a PPV of 10% which is based on the prevalence of ovarian cancer (1 in 2500 postmenopausal women).

Response to comment 1:
We appreciate the thoughtful comments of the reviewer. We amended the statements as “ The search for effective screening methods continues. An acceptable screening method for detecting early-stage ovarian cancer requires a sensitivity greater than 75%, and a specificity of at least 99.6% to achieve a positive predictive value (PPV) of at least 10% (12).” in 1.Introduction section, page 2, lines 59-62.

Comment 2:

In the section 1 introduction, a brief summary at the end about the motivation of this work can improve the work.

Response to comment 2:

We have added a brief summary “The discussion below focuses on recent advances in biomarker development for early detection of ovarian cancer, including two current Food and Drug Administration (FDA)-approved markers, algorithms or indexes, and ongoing development of potential molecular biomarkers.” at the end of 1. Introduction section, page 2, lines 62-65.

Comment 3:

Line 65. ‘Useful’ can be replaced by ‘diagnostic’.

Response to Comment 3:
We have replaced ‘Useful’ by ‘diagnostic’. “Other than CA125, the study found no markers that provided diagnostic discriminatory information when ovarian cancer was detected more than 9 months after blood draw (16).” in lines 78-80.

Comment 4:

Line 68. ‘usefulness’ can be replaced by ‘sensitivity or accuracy’.

Response to Comment 4:
We have replaced ‘usefulness‘ by ‘accuracy‘. “However, the accuracy of CA125 for detecting early-stage ovarian cancer is limited; only 50% of early-stage patients have elevated CA125 levels, leading to low sensitivity (50–62%) for detecting early stage ovarian cancer.” in lines 82-84.

Comment 5:

In section 2.2, line 158, the statement of HE4 it not complete or even not accurate. Please check works such as https://www.mdpi.com/1422-0067/14/11/22655/htm and https://www.ncbi.nlm.nih.gov/pmc/articles/PMC5928211/. In a brief, Human epididymis protein 4 (HE4) was first identified in the epithelium of the distal epididymis, belonged to the family of whey acidic four-disulfide core (WFDC) proteins. HE4 is a endogenous peptide protease inhibitors. HE4 is also found to be involved in the innate immune response of epithelial tissues, including ovarian epithelial cells.

Response to Comment 5:

We have amended the text accordingly in section 2.2, line 187-189. “HE4 is a member of whey acidic four-disulfide core (WFDC) protein family that was originally identified in the epithelium of the distal epididymis (41). It is a peptide protease inhibitor involved in the innate immune response of epithelial tissues (42, 43).”

Comment 6:

This work is a review article, however, authors incorrectly used the phrase ‘we found’ as line 170, 221, or ‘our study’ as line 230. Authors need to rephrase them appropriately.

Response to Comment 6:.

We rephrased them appropriately as follows.

Lines 198-200: Furthermore, Chan et al. found that HE4 showed better sensitivity in mucinous tumors, but is not strongly expressed in clear cell carcinoma (20).

Lines 298-300: Chan et al. found that ROMA had an 83% accuracy in diagnosing early-stage disease (20) and its sensitivity was higher in post-menopausal women than in pre-menopausal women (82.5–90.8% vs. 53.3–72.7%).

Lines 307-314: Chan et al. further found that ROMA showed improved specificity and PPV (34.69% vs 16.8%), but similar specificity and PPV when compared with CA125 for prediction of ovarian cancer in Asian women with a pelvic mass.

Lines 323-328: Chan et al.  have evaluated the IOTA with subjective assessment using expert ultrasound, RMI, and ROMA for assessing the nature of a pelvic mass (71). Chan et al. have investigated whether ROMA could replace expert ultrasound when the IOTA results are inconclusive, showing that expert ultrasound was more sensitive than ROMA for diagnosing ovarian malignancy in such cases, without significant differences in the specificity or accuracy (71).

Comment 7:

In section 3.1, authors only mentioned RMI briefly. However, several improved versions of RMI such as RMI2, 3 and 4 have been reported and used in the clinic https://ovarianresearch.biomedcentral.com/articles/10.1186/s13048-020-00643-6. It would be more instructive if authors can discuss them more deeply and with more details.

Response to Comment 7:

We have amended section 3.1 in lines 259-277 as follows.

“3.1. Risk of malignancy index (RMI) assay

Due to the limited efficacy of single serum biomarkers, many researchers have tried to combine several indexes to improve the performance biomarker applications. In 1990, Jacobset al. have established the RMI by multiplying ultrasound results (U), CA125 levels, and menopausal status (M) to predict the risk of ovarian malignancy (RMI = U x M x CA125) (52). Using the cut-off value of 200, the RMI demonstrates increased sensitivity (71–88%) and specificity (74–92%) compared to assessment of CA125 levels alone (53).

Later, Tingulstad et al. have developed RMI 2 (sensitivity at 71% and specificity at 96%) and RMI 3 (sensitivity at 71% and specificity at 92%), and Yamamoto et al. have developed RMI 4 (sensitivity at 86.8% and specificity at 91%). These three new versions employ modified scoring of U and M parameters. In addition, RMI 4 also takes tumor size into consideration (5456). The cut-off values with the best differentiation points are 200 for RMI 1–3 and 450 for RMI 4.

Many studies have been conducted to compare the efficacy of these four RMI versions. A systematic review has reported that RMI 1 had the highest accuracy among RMI 1–3 (57). The NICE guidelines based on this study recommended to use the RMI 1 score to manage suspected ovarian malignancy (58). However, a recent study has shown that despite the higher sensitivity in RMI 2 and 4, as well as the higher specificity in RMI 1, no significant differences in AUC were found among them (59).”

Comment 8:

At the conclusion section, line 391, identifying biomarkers is not a problem. Authors need to rephrase it more correctly, such as ‘unmet medical need or goal’.

Response to Comment 8:

We have amended the sentence accordingly in lines 702-703 as follows. “Identifying more accurate molecular biomarkers for early detection of ovarian cancer remains an important unmet medical need.”